# Comparison of the Effects of High Pressure Processing, Pasteurization and High Temperature Short Time on the Physicochemical Attributes, Nutritional Quality, Aroma Profile and Sensory Characteristics of Passion Fruit Purée

**DOI:** 10.3390/foods11050632

**Published:** 2022-02-22

**Authors:** Huihui Niu, Lei Yuan, Hengle Zhou, Yurou Yun, Jian Li, Jun Tian, Kui Zhong, Linyan Zhou

**Affiliations:** 1Faculty of Food Science and Engineering, Kunming University of Science and Technology, Kunming 650500, China; niuhuihui0606@163.com (H.N.); yl1576432@163.com (L.Y.); zhl219602@163.com (H.Z.); yyr020266@163.com (Y.Y.); lijianfood@foxmail.com (J.L.); tj7920921@163.com (J.T.); 2China National Institute of Standardization, Beijing 100191, China; zhongkui@cnis.gov.cn

**Keywords:** high temperature short time (HTST), pasteurization (PT), antioxidants capacity, aroma, sensory evaluation

## Abstract

The study investigated the effects of high-pressure processing (HPP) (600 MPa/5 min), pasteurization (PT) (85 °C/30 s), and high-temperature short time (HTST) (110 °C/8.6 s) on physicochemical parameters (sugar, acid, pH, TSS), sensory-related attributes (color, aroma compounds), antioxidants (phenolics, vitamin C, carotenoids, antioxidant capacity), and sensory attributes of yellow passion fruit purée (PFP). Compared to the PT and HTST, HPP obtained the PFP with better color, sugar, and organic acid profiles. Although PT was equally effective preservation of antioxidants and antioxidant capacity of PFP compared to HPP, high temperature inevitable resulted in the greater degradation of the aroma profile. The amounts of esters, alcohols, and hydrocarbon in PFP were significantly increased by 11.3%, 21.3%, and 30.0% after HPP, respectively. All samples were evaluated by a panel comprising 30 panelists according to standard QDA (quantitative descriptive analysis) procedure, and the result showed that HPP-treated PFP was rated the highest overall intensity score with 7.06 for its sensory attributes, followed by control (6.96), HTST (6.17), and PT (6.16). Thus, HPP is a suitable alternative technology for achieving the good sensory quality of PFP without compromising their nutritional properties.

## 1. Introduction

Passion fruit is one of the most popular species in the Passiflora family and is widely planted in the subtropical and tropical regions of Asia, America, Africa, and Australia [1]. Recently, the interest of researchers and producers has been stimulated by passion fruit due to its good nutritional characteristics and typical sensory attributes. The yellow passion fruits are a powerful source of antioxidants and bioactive compounds [2], being rich in vitamin C, carotenoids, and phenolic compounds [3]. Some phenolic compounds have been characterized in passion fruit products, and the major phenolic compounds were phenolic acids and flavonoids, such as quercetin, rutin neochlorogenic acid, vitexin, isoquercetin, and ferulic acid [4,5]. These compounds exhibit good antioxidant capacity, which can neutralize the free radicals present in many pathological processes, decrease the risk of cardiovascular diseases, and act as carcinogenesis and mutagenesis inhibitors [3].

The passion fruits are usually commercialized in the form of fresh fruit, purée, juice, or concentrated pulp, which are appreciated for their unique exotic aroma and color. Among them, passion fruit purée (PFP) exhibits an increasing market value because it is a convenient food product or ingredient with a natural fresh appearance and aroma. The aroma of passion fruit contributes to the great popularity of this fruit and directly affects the sensory quality of fresh passion fruit and its products, which arise from a complex combination of several secondary metabolites, such as formaldehyde, alcohols, ketones, esters, and terpenes [6]. Porto-Figueira et al. [7] reported that esters were the dominant aroma compounds in nine species of passion fruit, including hexyl hexanoate (6–31%), methyl hexanoate (14–75%), ethyl hexanoate (12–53%), and hexyl butanoate (11–26%).

Thermal pasteurization is one of the most important procedures in juice and purée processing and has been widely used in recent years to improve food safety and extend the shelf life of juice products [8]. Heat, however, inevitably leads to quality deterioration in foods by producing undesirable changes in sensory characteristics and decreasing nutritional properties [9]. Aroma compounds of passion fruit products have exhibited extreme sensitivity to thermal temperatures. Sandi et al. [10] found that about 50% of the esters were lost in the passion fruit juice after pasteurization (PT) at 80 °C for 60 s, as compared to the fresh juice. Moreover, significant changes in nutritional and functional compounds were usually reported for fruit products after thermal PT. For example, vitamin C is the relevant nutrient in yellow passion fruit within a range of 0.16–0.20 g/kg [11], which is thermoplastic and easily degraded after thermal PT. It was also reported that vitamin C in kiwifruit was significantly reduced by 38.39% after thermal treatment (110 °C/8.6 s) [12].

High-pressure processing (HPP) is a non-thermal technology that uses pressure to inactivate microorganisms and enzymes, while also reducing the damage to the nutrients and aroma [13]. Numerous studies have shown that HPP could better preserve the aroma and nutritional compounds in fruit juice and purée. For example, although β-myrcene, d-limonene, and 4-carene in the HPP treated juice were significantly lower than those in the fresh mango juice, the sensory test scores indicated that the juice after HPP at 600 MPa and 25 °C for 5 min had higher similarity with the fresh than the PT samples [14]. Laboissière et al. [9] showed that the fresh and the HPP (300 MPa/5 min)-treated yellow passion fruit juice were mostly well-differentiated from all commercial PT samples with high similarity in sensory attributes. However, limited information was reported on the changes in the aroma profile of PFP after HPP, PT, and HTST. Meanwhile, the previous study has shown that HPP exhibited better preservation effects on nutritional compounds, such as phenolics, carotenoids, and vitamin C in kiwifruit juice, pineapple juice, and mango juice [15].

In the present study, we comprehensively evaluated and compared the physicochemical parameters (sugar, acid, pH, TSS), sensory-related attributes (color, aroma compounds), antioxidants (phenolic, vitamin C, carotenoids, antioxidant capacity), and sensory attributes of yellow PFP treated with HPP, PT, and HTST. The knowledge obtained will help develop PFP as a new food ingredient, to improve the sensory attributes, consumer acceptability, and functional characteristics of these products.

## 2. Materials and Methods

### 2.1. Materials

#### 2.1.1. Passion Fruit Purée (PFP) Preparation and Processing

The yellow passion fruits were purchased from a local passion fruit planting orchard (Dehong, Yunnan Province) and stored in a refrigerator at 4 °C for further experiments.

The passion fruits were opened, the pulp was separated from the seeds to make purée using a home juicer, and the resulting homogenization was filtered with two layers of nylon gauze to remove residue [9]. The obtained purée was temporarily stored at 4 °C until processing.

#### 2.1.2. High-Pressure Processing (HPP) Treatment

In the food industry, HPP at 500–600 MPa yielded food products with good quality and safety [16]. The obtained purée was divided into 250 mL PET bottles and treated with the HPP. HPP was carried out in ultrahigh-pressure equipment (SHPP-DZ-600, Sanshuihe Tech. Co., Ltd., Taiyuan, Shanxi, China), which had a 2 L pressure vessel with a diameter of 90 mm and a height of 400 mm. The initial temperature of water in the processing chamber was 20 °C. The purée was pressurized at 600 MPa for 5 min, with a pressure increase rate of approximately 7.5 MPa/s, and the pressure release time was <3 s after the HPP treatment. The duration of treatment did not include come-up and release time.

#### 2.1.3. Thermal Pasteurization Treatment

Thermal pasteurization was conducted in a multipurpose ultrahigh temperature UHT sterilization unit (ST-20, Shanghai Sunyi Tech. Co., Ltd., Shanghai, China). Two levels of processing intensity were selected: pasteurization (PT) (85 °C/30 s), and high-temperature short time (HTST) (110 °C/8.6 s). To effectively destroy pathogens and inactivate endogenous enzymes, purée was preheated to 65 °C and pasteurized at 85 °C for 30 s [17], and preheated to 95 °C and pasteurized at 110 °C for 8.6 s [18], respectively. The preheating tube for PT and HTST was the same and the length was fixed (910 mm), and the preheating time for both was about 30–40 s. The duration of treatment did not include preheating time.

### 2.2. Microbial Analysis

Microbial analyses were performed for HPP, PT, and HTST treated and untreated samples. One milliliter of purée was diluted (1:10 *w*/*w*) in sterile saline solution. The plate count agar was used for counting the total aerobic bacteria (TAB) after incubation at 36 ± 1 °C for 48 ± 2 h. The number of yeast and mold (Y&M) samples were detected after incubation in rose bengal agar at 28 ± 1 °C for 5 d. Then, the microorganism numbers of the samples were enumerated as a log of CFU/mL.

### 2.3. Enzyme Activity Analysis

The extraction of polyphenol oxidase (PPO) and peroxidase (POD) and analysis of activity were performed using the method described by Yi et al. [19] with minor modifications. Briefly, 3 mL sample was mixed with 3 mL of solution composed of 4% (*w*/*v*) insoluble PVPP, 1 M NaCl, and 1% (*w*/*v*) Triton X-100 in 0.2 M sodium phosphate with a final pH of 6.5, and centrifuged at 14,000× *g* and 4 °C for 30 min. More details of the methodology can be found in Appendix A.

### 2.4. Total Soluble Solids (TSS), Total Sugar (TS) and Sugar Profile Analysis

TSS was determined with a Brix refractometer (TD-45, Beijing Jinkelida Electronic Technology Co., Ltd., Beijing, China) at 25 ± 1 °C and the results were expressed as °Brix. TS was determined by Fehling reagent titration method [20]. The results were expressed as standard glucose content (g/100 g).

The sugar profile was analyzed by high-performance liquid chromatography (HPLC) using the procedure of Pham et al. [21] with some modifications. For sugar extraction, 250 µL Carrez I (0.41 mol/L K_4_ [Fe(CN)_6_]) and 250 µL Carrez II (1.86 mol/L ZnSO_4_) were added to 5 mL purée. The mixture was homogenized with a vortex mixer for 3 min. After standing at room temperature for 30 min, the mixture was centrifuged (9570× *g*, 4 °C) for 15 min. The supernatant was diluted (1:9) in HPLC-grade water and filtered through a 0.45 µm nylon membrane for determination of individual sugar by using HPLC (G1315B; Agilent, Santa Clara, CA, USA) with evaporative light scattering detection (ELSD, G4260B, Agilent, Santa Clara, CA, USA). More details of the methodology can be found in Appendix A.

### 2.5. pH, Titratable Acid (TA) and Organic Acid Analysis

The determination of pH value was carried out by an Orion 868 pH meter (FE28-standard, Mettler Toledo, Zurich, Switzerland) at 20 ± 1 °C. TA was determined by titrating with standardized 0.1 mol/L NaOH, reaching pH 8.1 by an automatic potentiometric titrator (907 Titrando, Metrohm AG, Herisau, Switzerland) [22]. TA was expressed as citric acid equivalents were the predominant acid in passion fruit, as reported in a previous study [3]. *TA* content was calculated using Equation (1) [23].
(1)$$TA\left(g/100\;g\right)=(C\times V_{2}\times K\times V_{0}\times 200)/(V_{1}\times m)$$
where *C* is the NaOH concentration (0.1 mol/L), m (g) is the weight of purée, *V*_0_ (mL) is the total volume of purée, *V*_1_ (mL) is the purée used, *V*_2_ (mL) is the volume of NaOH used, and *K* is the conversion factor of citric acid (0.07).

The extraction procedure for organic acids was the same as individual sugars described in 2.4. Extraction solution was diluted (1:4) using HPLC grade water and filtered through a 0.45 µm syringe filter. Following the procedure of Wibowo et al. [17], the organic acid profile was analyzed by using a reversed-phase HPLC (1260 Infinity, Agilent, Santa Clara, CA, USA) equipped with a Prevail Organic Acid column (250 mm × 4.6 mm, 5 μm particle size, Avantor, Radnor Township, PA, USA). More details of the methodology can be found in Appendix A.

### 2.6. Color Analysis

The color of PFP was determined by using the CIE L*a*b* system and a colorimeter (Agera, Hunter Associate Laboratory, Inc., Fairfax, USA) with D65 illuminant and 10° observer angle. Total color difference (Δ*E*) was calculated by Equation (2) [24].
(2)$$\Delta E=\sqrt{{\left(L^{*}-L_{0}{}^{*}\right)}^{2}+{\left(a^{*}-a_{0}{}^{*}\right)}^{2}+{\left(b^{*}-b_{0}{}^{*}\right)}^{2}}$$

The L*, a* and b* signify the measured brightness value, redness value, and yellowness value of the PPF by different treatment, respectively, and the subscript ‘0’ stood for untreated samples.

### 2.7. Aroma Compounds Analysis

The extraction of aroma compounds was following the method described by Pan et al. [14] using divinylbenzene/carboxen/polydimethylsiloxane solid-phase microextraction (SPME). An internal standard method was used to quantify the identified aromas. PFP (5 mL) was transferred into a headspace bottle containing 1.8 g NaCl and 1 μL of Butyl 2-methylbutyrate (100 μL/L, as internal standard). The bottle was sealed by parafilm septum and equilibrated at 40 °C for 5 min. Then, aroma compounds extracted were determined by gas chromatography-mass spectroscopy (GC-MS). More details of the methodology can be found in Appendix A. The quantification of aroma compounds was performed using Butyl 2-methylbutyrate as an internal standard.

### 2.8. Total Phenolics Content (TPC), Total Flavonoids Content (TFC) and Antioxidant Capacity Analysis

According to the method of Wang et al. [25] with slight modifications, antioxidants were extracted with PFP diluted by water/methanol (1:4) with the ratio of 1:3 (*v*/*v*), sonicated for 20 min, and centrifuged at 9000× *g* and 4 °C for 5 min. The obtained supernatant was used for the TPC, TFC, and antioxidant capacity determination on a microplate reader (EPOCG/2, BioTek, Winooski, VT, USA), according to Wang et al. [25].

#### 2.8.1. TPC Analysis

The determination of TPC was carried out by using the classical Folin–Ciocalteau assay with slight modifications. A total of 50 µL extract was mixed with the 10-fold-diluted Folin–Ciocalteu reagent (500 µL) and 450 µL Na_2_CO_3_ (0.71 mol/L), then reacted at room temperature in the dark for 1 h. A 200 µL solution was pipetted into the microplate and measured at 765 nm. The results were expressed in gallic acid equivalent per liter of purée (mg GAE/L). At 6 min, 100 µL of 1.0 mol/L NaOH was added and mixed. The absorbance was determined at 510 nm.

#### 2.8.2. TFC Analysis

TFC was determined by AlCl_3_ colorimetry with some modifications. Firstly, 100 µL of extraction solution and 5 µL of 0.72 mol/L NaNO_2_ were added to the microplate, and 0.37 mol/L AlCl_3_ (15 µL) was added 5 min later. After incubation for 30 min at room temperature, the absorbance was measured at 510 nm. The result was expressed in rutin equivalent per liter of purée (mg RE/L).

#### 2.8.3. Phenolics Analysis

Six independent repetitions were executed for the extraction and analyzed by a Thermo Fisher Ultimate 3000 UHPLC system equipped with a Q-Exactive Orbitrap mass spectrometer (Thermo Fisher Scientific, Bremen, Germany). More details of the methodology can be found in Appendix A.

#### 2.8.4. Antioxidant Capacity Analysis

##### DPPH Assay

DPPH reaction solution was prepared by adding water/methanol (1:4) to adjust its absorbance to 0.90 ± 0.05 at 517 nm. Forty microliters of diluted samples (1:10) and 160 µL DPPH reaction solution was added to the microplate, then reacted for 30 min in the dark. Absorbance value was measured at 517 nm. The results were expressed as mmol Trolox equivalent (TE)/L of purée.

##### ABTS^•+^ Assay

The ABTS^•+^ solution was diluted with methanol to an absorbance of 0.70 ± 0.02 at 734 nm. The ABTS^•+^ solution was produced by reacting 7 mmol/L ABTS stock solution with 2.45 mM K_2_S_2_O_8_ and kept in dark at room temperature for 12–16 h before use. Diluted methanolic extract (1:9) was mixed with ABTS^•+^ solution (A_734_ = 0.70 ± 0.02) and incubated for 6 min at room temperature, then the absorbance was measured at 734 nm. Results were expressed in mg Trolox equivalent (TE)/L of purée.

### 2.9. Vitamin C Analysis

Total vitamin C includes ascorbic acid (AA) and dehydroascorbic acid (DHAA). According to Cao et al. [23], 5 mL of purée was mixed with 20 mL of extraction solution (0.13 mol/L HPO_3_ and 0.08 mol/L CH_3_COOH, pH 2.0), and centrifuged at 10,000× *g* for 30 min at 4 °C. The obtained extract was divided into two parts: one was used for AA analysis and the other was used for vitamin C analysis, which was further analyzed by HPLC (G1315B; Agilent, Santa Clara, CA, USA). The details of the methodology can be found in Appendix A.

### 2.10. Carotenoids Analysis

The extraction of carotenoids was according to the method of Giuffrida et al. [26], with some modifications. PFP (10 mL) were extracted with 5 mL of solution (CH_3_OH/EtAc/CH_2_Cl_2_, 25:25:50, *v*/*v*/*v*, containing 0.005 mol/L of BHT), stirred for 5 min, and placed in an ultrasound bath for 5 min to enhance extraction. The mixture was centrifuged at 17,000× *g* at 4 °C for 5 min. These operations were repeated until color exhaustion was found with extracting the solvent. The organic phase containing carotenoids was separated and pooled. Finally, the organic phase was concentrated to dryness by rotary evaporation at 30 °C and analyzed by HPLC (G1315B; Agilent, Santa Clara, CA, USA). More details of the methodology can be found in Appendix A.

### 2.11. Sensory Evaluation Analysis

Samples after different treatments were evaluated by a panel comprising 30 panelists by following standard QDA (quantitative descriptive analysis) procedures [9]. The samples were coded using three random numbers and presented to the assessors at room temperature and under white lightning in capped glass bottles. Water was provided for the panel to rinse their palates between samples. Sixteen attributes for the characterization of purée were selected for sensory panel (Appendix A). Before scoring, untreated PFP was prepared for reference sample as a standard. A nine points scale (0 = no attribute, 9 = very intense) was utilized to evaluate the intensity of each descriptor for sensory properties. The overall intensity score of sensory evaluation was taken as the average intensity score of each index by the evaluator.

### 2.12. Statistical Analysis

Statistical analysis was performed using Origin 8.0 (OriginLab, Inc., Northampton, MA, USA) and SPSS 20.0, and results are expressed as mean ± SD. A one-way analysis of variance (ANOVA) was used to perform Tukey’s significant difference test, and *p* < 0.05 is significant. Variable importance in projection (VIP) coefficients were calculated to select discriminant compounds, and those values with the largest |VIP| > 1 were selected for principal component analysis (PCA) by bioinformaticsa free online data processing software. Metaboanalyst, a free online data processing software, was used for heatmap analysis.

## 3. Results and Discussion

### 3.1. Microbial and Enzyme

As shown in Table 1, the initial levels of the total aerobic bacteria (TAB) and yeasts and molds (Y&M) were 3.89 ± 0.30 log CFU/mL and 2.16 ± 0.11 log CFU/mL in control passion fruit purée (PFP), respectively. No TAB and Y&M counts were detected in PFP after all processing technologies were investigated, including high-pressure processing (HPP), pasteurization (PT), and high-temperature short time (HTST). A similar result was also found by Hu et al. [13], where TAB and Y&M were both completely inactivated in jabuticaba juice after TP (90 °C/30 s) and HPP (600 MPa/5 min) treatments [13]. As a highly acidic juice (pH: 3.02), it was expected that HPH could achieve an effective sterilization effect for PFP. PT and HTST led to the complete inactivation of polyphenol oxidase (PPO), whereas the relative activity of 30.77 ± 10.88% was found for PPO after HPP. A different situation was found for peroxidase (POD) that PT and HTST caused, decreasing around 44–46%, while a higher relative activity of 85.42 ± 1.72% was detected for the HPP-treated sample. That was to say, POD proved more stable towards heat and pressure than PPO. It has been reported that high pressure affected the enzyme conformation through compaction and change in molar volume, and accompaniment by temperature elevation during HPP resulted in the loss of enzyme functionality [27].

### 3.2. Sugars and Organic Acids

As can be seen in Table 2, the PFP of TSS, total sugars (TS), pH, and total acids (TA) in control were 12.46 ± 0.04 °Brix, 11.89 ± 0.30 g/100 g, 3.02 ± 0.03, and 5.80 ± 0.12 g/100 g, respectively. Processing technologies, including HPP, PT, and HTST, all have little effect (*p* > 0.05) on those values of the PFP. Similar results were also found by Yi et al. [19] and Wu et al. [28], where the TSS, TS, pH, and TA in cloudy apple juice and pineapple fruit juice after HPP (600 MPa/3 min and 500 MPa/10 min) and thermal pasteurization (85 °C/5 min and 95 °C/5 min) were not significantly changed. Sugars and organic acid profiles are inherently responsible for the sweetness and sourness of the fruit products, respectively, playing a decisive role in the sensory properties and acceptability of fruit products [17,29]. The main sugar in control PFP was sucrose, accounting for more than 57.5% in sugar, followed by glucose (22.1%) and fructose (20.5%), confirming the 2:1:1 ratio usually mentioned in literature [29]. HPP and PT showed no significant influence on the sugar profile of PFP, but HTST significantly changed those values. The content of sucrose was significantly increased by 13.6% after HTST, while the fructose and glucose were significantly decreased by 13.6% and 16.6%, respectively. The possible explanation for this phenomenon in our study could be that HTST significantly reduced the activity of acid invertase, which inhibited the conversion of sucrose to fructose and glucose [30]. A similar result was also found by Wibowo et al. [17], where the sucrose content in apple juice was increased by 4.3% after PT (85 °C/30 s), while the fructose and glucose were decreased by 18.0% and 9.3%, respectively.

A total of six organic acids were detected in PFP, including oxalic acid, malic acid, lactic acid, acetic acid, citric acid, and quinic acid. Citric acid, with a content of 25.90 ± 1.82 mg/mL, was the main organic acid in control PFP, accounting for 81.8% in the total organic acids and consistent with a proportion of 85.3% reported for yellow passion fruit juice in the Xie et al. [5] study. The contents of individual organic acids were all not changed by HPP, indicating that HPP did not alter the sourness of PFP. Significant decreases in oxalic acid and lactic acid content were found in PT and HTST-treated PFP, while malic acid, acetic acid, citric acid, and quinic acid were not changed by PT and HTST. Malic acid, acetic acid, citric acid, and quinic acid in orange juice and cloudy apple juice also showed stability towards thermal pasteurization (72 °C/20 s and 85 °C/30 s) [17,29]. A similar result was also found by Wibowo et al. [17], where the sucrose contents in apple juice were increased by 4.3% after thermal pasteurization (85 °C/30 s) while the fructose and glucose were decreased by 18.0% and 9.3%.

### 3.3. Color

As shown in Table 3, the initial L*, a*, and b* values of the control PFP were 59.21 ± 1.25, 11.94 ± 0.32, and 62.70 ± 0.45, respectively. There was no significant change in color parameters between the control and the HPP-treated samples. It is well known that HPP has little effect on the covalent bond of the equimolar mass of the color compound, thus it can protect the color well [31]. In contrast, PT and HTST significantly increased L* value by 4.6% and 5.3%, respectively. The ΔE value of PFP treated by HPP was 2.26 ± 0.06, and it significantly increased to 3.01 ± 0.39 and 3.06 ± 0.08 after PT and HTST, respectively—indicating HPP better preserved the color of PFP as compared with PT and HTST. Furthermore, PT and HTST caused obvious color changes in PFP with ΔE > 3, indicating that the color changes induced by thermal pasteurization could be perceived by inexperienced observers [28]. It was found that the PPO in PFP was completely inactivated by PT and HTST; meanwhile, the relative activity of POD in PT-and HTST-treated samples was significantly lower than that of HPP-treated sample. Thus, it was deduced that the color changes in PFP after PT and HTST treatment was mainly induced by non-enzymatic browning, such as the Maillard reaction and pigment destruction [19].

### 3.4. Aroma Profile

A total of 51 aroma compounds were identified in PFP, including 23 esters, 12 alcohols, 7 hydrocarbons (6 terpenes and 1 hydrocarbon), 5 ketones, 3 acids, and 1 aldehyde (Appendix A). Most of the aroma compounds detected in PFP had a pleasant aroma of “fruity, floral and winey”, which have been reported in passion fruit products by other researchers [32,33]. There were eight compounds identified in our study that were not reported for passion fruit products in known literature, namely, prenylacetone (38.73 ± 2.05 µg/L), sulcato (11.52 ± 0.84 µg/L), iso-mentone (18.15 ± 2.48µg/L), decanal (8.60 ± 1.30 µg/L), theaspirane (15.40 ± 0.90 µg/L), nerylacetone (18.13 ± 2.13 µg/L), 1-octadecano (14.75 ± 2.26 µg/L), and octadecanoic acid (11.18 ± 0.64 µg/L).

Figure 1A showed that esters were the most abundant aroma compounds detected in control PFP, accounting for 57.5%, which mainly exhibited fruity characteristics with relatively high sensory thresholds [17]. The ethyl butanoate was the most abundant ester in PFP with a content of 3133.51 ± 361.05 µg/L, followed by ethyl hexanoate (1051.59 ± 73.34 µg/L). Alcohols were the second largest group identified in PFP, accounting for 31.4% in the overall aroma compounds, which mainly contributed to the odors of winey, green, flowery, fruity, and sweet [34]. Among these, the highest value of 2487.15 ± 341.76 µg/L was found for 1-hexanol, followed by isoamylol with a content of 354.46 ± 14.33 µg/L. Acids were accounting for 3.5% of the aroma compounds in PFP, contributing to the green oily odorless and mild fatty waxy odors. There were also other small portions of aroma compounds detected in PFP, including ketones and aldehyde.

As compared with control PFP, the amounts of esters, alcohols, and hydrocarbon in the HPP-treated sample were significantly boosted by 11.3%, 21.3%, and 30.0%, respectively, while those values in the PT- and HTST-treated samples were significantly decreased by 40.7–48.0%, 80.3–81.8%, and 66.7–71.3%, respectively. Previous results also reported that HPP could better preserve the aroma compounds of fruit products, such as apple juice and mango juice [14,19].

Figure 1B shows that all samples were clearly divided into two clusters. Cluster 1 included the control and HPP-treated sample, while cluster 2 included PT- and HTST-treated samples. The aroma profile of HPP-treated PFP was closer to that of the control, while PT and HTST greatly reduced most aroma compounds of the purée. It was found that HPP increased some esters and alcohols. Sulcatol, ethyl-hydroxybutyrate, ethyl 3-hydroxyhexanoate, and (E)-3-hexenyl butyrate in HPP-treated PFP were significantly increased by 268.9%, 135.4%, 58.2%, and 42.5%, respectively. In addition, ethyl butanoate and ethyl hexanoate, which were the two most abundant compounds in PFP, were significantly increased by 38.6% and 9.6% after HPP, respectively. The increase of esters would increase the pleasant fruity aroma of the juice [17], that was to say, HPP could improve the overall aroma profile of PFP. HPP could indirectly alter the content of some aroma compounds by enhancing enzymatic and chemical reactions, which could lead to desirable changes in the overall aroma profile [35]. Hexyl octylate, ethyl propinoate, and citronellol were decreased after PT and completely lost after HTST. It was indicated that these compounds were sensitive to higher temperatures. The loss of esters and citronellol might cause the weakening of fresh, fruity, or flora and grassy aroma compared with the control sample.

Octadecanoic acid, 1-heptanol, nerylacetone, linalool, and linalool oxide were significantly increased by 647.8%, 201.2%, 97.9%, 76.4%, and 58.5% after HTST, respectively. The formation of these potentially temperature-induced compounds could be linked to the Maillard reaction and oxidative reactions (e.g., carotenoids and unsaturated fatty acid degradation) [36]. Some increasing aroma compounds possibly generated some unpopular aroma. For example, octadecanoic acid and 1-heptanol were identified as contributors to the fatty and musty aroma in PFP, which have been reported to directly impact the sensory quality of sugarcane juice [34].

### 3.5. Antioxidants and Antioxidant Capacity

Table 4 shows that most antioxidants in PFP, including total phenolics content (TPC), total flavonoids content (TFC), and vitamin C, were better preserved by HPP and PT as compared to HTST. Correspondingly, the antioxidant capacity determined by DPPH and ABTS^•+^ in PFP after HPP and PT was significantly higher than those values of HTST treated sample.

As compared to the control PFP, HPP did not change the contents of antioxidants, except that TPC was decreased by 8.3%. This was possibly explained by an increase in condensation reactions of the phenolic compounds in the PFP promoted by HPP [36]. The initial values of vitamin C, ascorbic acid (AA), and dehydroascorbic acid (DHAA) in the control PFP were 251.68 ± 1.03, 122.26 ± 2.86, and 129.42 ± 2.98 mg/mL, respectively, and these values were all not changed by HPP. Pressure has a low impact on covalent bonds and therefore does not directly damage small molecules, such as vitamin C [16]. PT did not change the content of vitamin C in PFP but resulted in a 42.2% decrease of AA and a 23.5% increase in DHAA. HTST induced similar changes with a greater loss of vitamin C and AA. Vitamin C has an extremely unstable nature and thus is greatly affected by temperature [16]. The main cause of vitamin C degradation is that AA can first be degraded to DHAA by oxidation reaction, and then DHAA can be hydrolyzed to 2,3-diketogulonic acid (DKG) before DKG is further oxidized to over 50 substances [37].

To better understand the individual phenolic compound changes in PFP after different technologies, the phenolic compounds were identified and quantified by using LC-MS. A total of 15 phenolic compounds were identified in control PFP (Appendix A). The main phenolic compounds detected in the control sample were caffeic acid hexoside (3496.78 ± 24.07 µg/L), phlorizin (1874.49 ± 39.11 µg/L), and rutin (1317.7 ± 168.89 µg/L). It was found that the phenolic compounds in passion fruit varied greatly, possibly due to the variety, place of origin, and the determination method and equipment used. Xie et al. [5] found that neochlorogenic acid was the major compound in two varieties of passion fruit juice from Guangdong, Fujian, Yunnan, and Guangxi Province of China regions, which ranged from 16.55 to 129.07 μg/mL.

As shown in Figure 2, the phenolic profiles of HPP- and PT-treated PFP were closer to that of the control, while HTST greatly reduced most phenolics in PFP. Protocatechuic acid and rutin slightly reduced after HPP, and the other 13 individual phenolic compounds were not significantly influenced by HPP. The contents of caffeic acid hexoside, astralagin, eriodictyol-7-*O*-glucoside, and luteolin-4-*O*-glucoside in PFP were significantly increased by PT, while the contents of protocatechuic acid, galloyl-glucoside, and *p*-coumaric acid were slightly decreased. The different stability of phenolics towards PT was caused by the different structures of these phenolics. Moreover, some phenolics might bind tightly with food substrates through covalent bonds, and PT would have limited ability to destroy these glycoside bonds [38]. HTST caused significant decreases in the phenolic content in PFP, except that eriodictyol-7-*O*-glucoside was significantly increased by 40.1% after HTST. The increase of glycoside content was possibly due to the breakdown of a cell wall structure and hydrolysis of polysaccharides induced by a high temperature (90 °C/60 s) [15]. A similar result was observed where quercetin glycosides in red raspberry juice were significantly increased by HTST-treated (110 °C/8.6 s) [39].

There were four carotenoids detected in PFP, namely, β-cryptoxanthin, zeaxanthin, β-carotene, and lycopene. The highest content of 45.36 ± 2.88 µg/mL was found for β-cryptoxanthin in fresh purée, which accounted for 46.8% of the total carotene. As compared to the control sample, HPP had no significant effect on β-carotene, lycopene, and total carotenoid contents in PFP, whereas zeaxanthin in HPP-treated PFP was increased by 56.8%, and correspondingly β-cryptoxanthin was decreased by 39.3%. It was reported that β-cryptoxanthin was (R)-isomer of β-carotene, which could be converted into zeaxanthin by the enzymatic action of the β-ring hydroxylase [40]. It was deduced that the HPP might promote the enzymatic reaction of β-ring hydroxylase in PFP, resulting in the conversion of β-cryptoxanthin to zeaxanthin. Zeaxanthin and lycopene in PFP proved more stable towards both PT and HTST, while β-carotene and β-cryptoxanthin were significantly decreased by 34.0% and 79.7% after HTST, respectively. Carotenoids showed greatly different amounts of stability towards different technologies, mainly due to their difference in molecular structure. It was also reported that lycopene in tomato juice was not significantly changed by PT (90 °C/90 s) and HPP (600 MPa/5 min) [41]. In addition, β-carotene in apricot nectar was stable under HPP at 300 MPa for 5 min and HTST at 110 °C for 8.6 s, while β-cryptoxanthin was significantly decreased by 25.7% and 13.5% after HPP and HTST, respectively [42].

### 3.6. Principal Component Analysis (PCA)

As shown in Figure 3, all samples were basically clustered according to the processing method. Clearly, there were significant differences in these indicators among PT, HTST, and HPP treatments. PC1 clearly separated PT and HTST-treated samples from control and HPP-treated ones, while PC2 allowed the discrimination of the PT sample from the HTST one. HPP-treated samples were positioned positively loading on PC2, demonstrating a substantial accumulation of aroma compounds, such as leaf alcohol, 1-hexanol, hexyl hydroxybutyrate, hexyl hexoate, and ethyl butanoate. This result suggested that the HPP was more favorable for these aroma releases. The PT- and HTST-treated samples showed a migration along PC2 from negative scores to positive scores, which was characterized by declines of aroma compounds (mainly esters and alcohols), antioxidants, antioxidant capacity, and increases of octadecanoic acid, 1-heptanol, and ethyl hexanoate. The markers that differentiated PT- and HTST-treated samples were eriodictyol-7-*O*-glucoside, octadecanoic acid, 1-heptanol, and ethyl hexanoate.

### 3.7. Sensory Evaluation

Sensory evaluations of PFP processed by HPP, PT, and HTST were shown in Figure 4. The high similarity between the sensory descriptive profiles of the control and HPP-treated PFP can be seen via the spider web plots. The PFP treated by HPP was rated with the highest overall intensity score at 7.06 ± 1.11 for its sensory attributes, followed by control (6.96 ± 1.16), HTST (6.17 ± 1.36), and PT (6.16 ± 1.29). These results indicated that the sensory attributes of the HPP-treated sample were closer to the control. As compared to the control PFP, the scores of purée glossiness, suspended particles, and sweet aroma of PFP after HPP were increased by 4.8%, 3.4%, and 3.5%, respectively, suggesting that HPP can obtain better color, aroma, and flavor compounds responsible for the sensory quality of PFP. Three sensory attributes of HTST-treated PFP, including glossiness (5.87 ± 0.99), cooked flavor (6.20 ± 1.30), and astringency (5.73 ± 1.12), were all rated the lowest, indicating higher temperatures could destroy the most sensory attributes of PFP. Results of sensory comparisons indicated that HPP retained better sensory properties of PFP, which were also supported by the results on physicochemical and sensory-related chemical indicators in this study.

## 4. Conclusions

In summary, HPP-treated PFP retained better color quality and more antioxidants of PFP compared to PT and HTST and had high similarity in the sensory descriptive profiles with the control sample. Although PT was equally effective at preservation within antioxidants, including TPC, TFC, vitamin C, β-carotene, zeaxanthin, and lycopene, as well as antioxidant capacity of PFP as compared to HPP, the high temperature inevitably resulted in the greater degradation of aroma profile and sensory descriptive profiles of PFP. The amounts of esters, alcohols, and hydrocarbon in the HPP-treated sample were especially significantly increased by 11.3%, 21.3%, and 30.0%, respectively, while most of the aroma compounds were significantly decreased by thermal pasteurization. Furthermore, the changes in sensory evaluation agreed with the changes in physicochemical characteristics such as color, pH, TSS, sugar, acid, and the aroma profile of PFP. Interestingly, zeaxanthin in HPP-treated PFP was increased by 56.8% and, correspondingly, β-cryptoxanthin was decreased by 39.3%, which was explained by the fact that HPP promoted the enzymatic action of the β-ring hydroxylase in PFP. Hence, HPP proved to be a promising preservation method of PFP, in contrast with thermal processing. The sensory analysis performed with consumers could provide more information related to HPP-treated purée acceptability in the future. Furthermore, the effect mechanism of HPP on antioxidants, such as carotenoids and phenolics, is worth studying based on the enzymatic reaction in future work.

## Figures and Tables

**Figure 1 foods-11-00632-f001:**
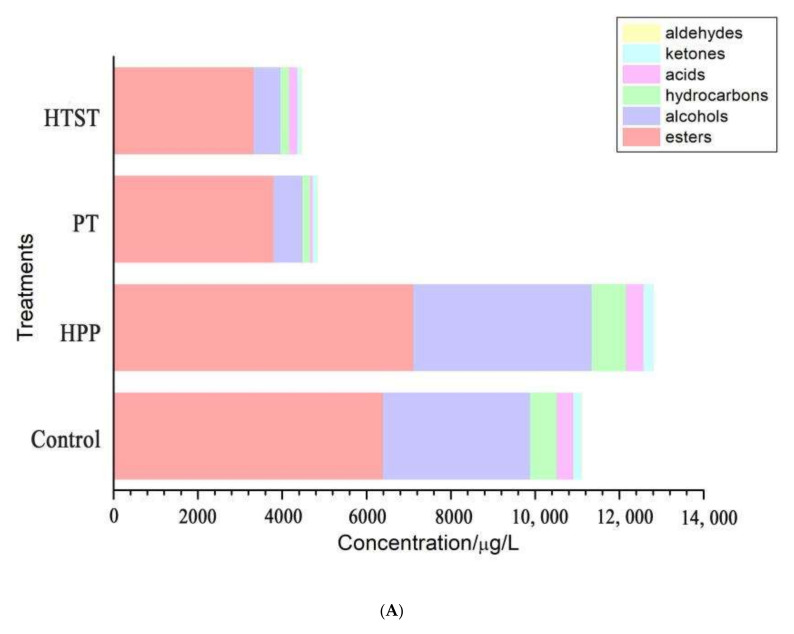
(**A**) Concentrations of major classes of aroma compounds of control, high-pressure processing (HPP), pasteurization (PT), and high-temperature short time (HTST)-treated passion fruit purée; (**B**) hierarchical clustering of the 51 quantified aroma compounds in control-, HPP-, PT-, and HTST-treated passion fruit purée.

**Figure 2 foods-11-00632-f002:**
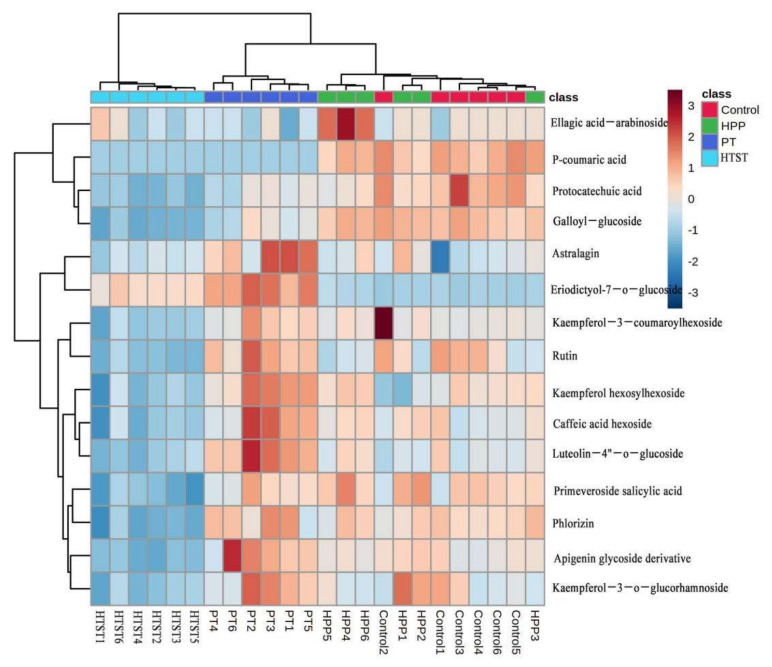
Hierarchical clustering of the 15 quantified phenolic compounds in control, high-pressure processing (HPP), pasteurization (PT), and high-temperature short time (HTST)-treated passion fruit purée.

**Figure 3 foods-11-00632-f003:**
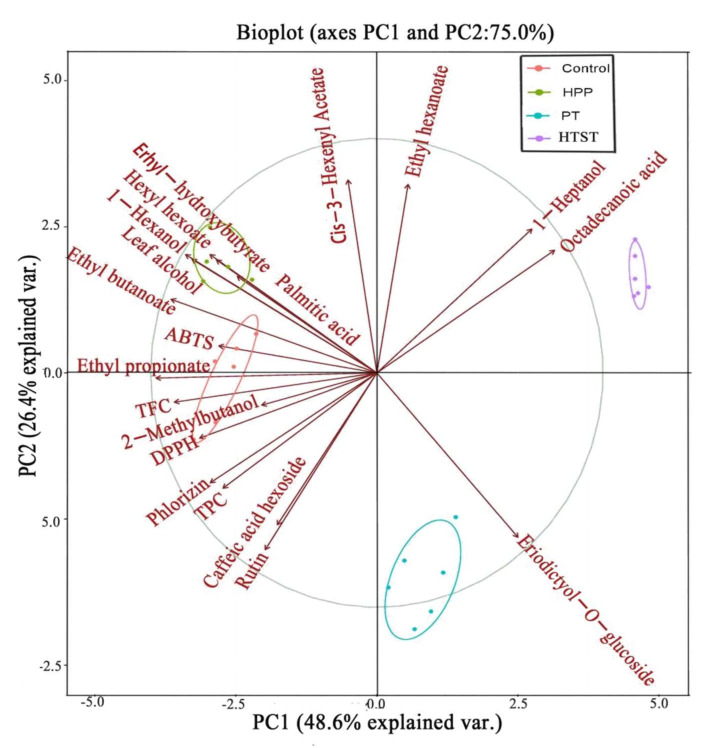
Principal component analysis (PCA) plot of the taste and related attributes, color and related attributes, aroma compounds, antioxidants and antioxidant capacity attributes in control, high-pressure processing (HPP), pasteurization (PT), and high-temperature short time (HTST) treatment passion fruit purée.

**Figure 4 foods-11-00632-f004:**
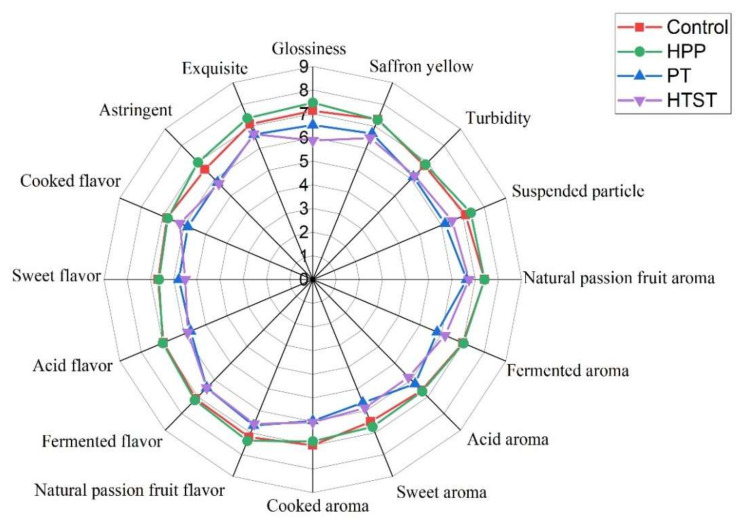
Spider plot of sensory attributes of control, high pressure processing (HPP), pasteurization (PT), and high-temperature short time (HTST))-treated passion fruit purée.

**Table 1 foods-11-00632-t001:** Effect of high-pressure processing (HPP), pasteurization (PT), and high-temperature short time (HTST) on the microbial (total aerobic bacteria (TAB), yeast and mold (Y&M)) and relative enzyme (polyphenol oxidase (PPO), peroxidase (POD)) activity of passion fruit purée.

Treatments	TAB (log CFU/mL)	Y&M (log CFU/mL)	PPO (%)	POD (%)
Control	3.89 ± 0.30	2.16 ± 0.11	100.00 ± 28.78 ^a^	100.00 ± 4.24 ^a^
HPP	nd	nd	30.77 ± 10.88 ^b^	85.42 ± 1.72 ^ab^
PT	nd	nd	0.00 ± 0.00 ^b^	55.56 ± 0.92 ^c^
HTST	nd	nd	0.00 ± 0.00 ^b^	53.86 ± 1.04 ^c^

Values are mean ± SD (*n* = 3). Means within columns with different letters (a–c) are significantly different (*p* < 0.05). nd: not detectable (<1 log CFU/mL).

**Table 2 foods-11-00632-t002:** Effect of high-pressure processing (HPP), pasteurization (PT), and high-temperature short time (HTST) on taste-related attributes (sugar and organic acid) of passion fruit purée.

Treatments	TSS (°Brix)	TS (g/100 g)	Reducing Sugars		Non-Reducing Sugar	pH	TA (g/100 g)	Organic Acids					
			Fructose (g/L)	Glucose (g/L)	Sucrose (g/L)			Oxalic Acid (mg/mL)	Malic Acid (mg/mL)	Lactic Acid (mg/mL)	Acetic Acid (mg/mL)	Citric acid (mg/mL)	Quinic Acid (mg/mL)
Control	12.46 ± 0.04 ^a^	11.89 ± 0.30 ^a^	112.36 ± 2.93 ^a^	121.16 ± 2.69 ^a^	315.56 ± 5.79 ^a^	3.02 ± 0.03 ^a^	5.80 ± 0.12 ^a^	0.56 ± 0.03 ^a^	2.53 ± 0.09 ^ab^	2.31 ± 0.21 ^a^	0.30 ± 0.02 ^a^	25.90 ± 1.82 ^a^	0.07 ± 0.01 ^a^
HPP	12.60 ± 0.08 ^a^	12.26 ± 0.61 ^a^	109.08 ± 6.37 ^ab^	120.94 ± 4.55 ^a^	315.67 ± 9.18 ^a^	3.04 ± 0.01 ^a^	5.73 ± 0.31 ^a^	0.59 ± 0.01 ^a^	2.37 ± 0.09 ^b^	1.00 ± 0.04 ^b^	0.31 ± 0.04 ^a^	24.85 ± 0.79 ^a^	0.06 ± 0.01 ^a^
PT	12.60 ± 0.21 ^a^	12.09 ± 0.47 ^a^	105.8 ± 11.32 ^ab^	116.54 ± 5.02 ^a^	324.95 ± 5.47 ^a^	3.04 ± 0.05 ^a^	5.31 ± 0.47 ^a^	0.26 ± 0.01 ^b^	2.36 ± 0.16 ^b^	1.25 ± 0.09 ^b^	0.31 ± 0.01 ^a^	25.40 ± 0.79 ^a^	0.07 ± 0.01 ^a^
HTST	12.56 ± 0.04 ^a^	11.85 ± 0.38 ^a^	86.59 ± 5.61 ^b^	101.09 ± 5.06 ^b^	358.44 ± 8.88 ^b^	3.04 ± 0.08 ^a^	6.01 ± 0.38 ^a^	0.29 ± 0.02 ^b^	2.84 ± 0.17 ^a^	1.49 ± 0.21 ^b^	0.49 ± 0.15 ^a^	29.78 ± 2.43 ^a^	0.08 ± 0.01 ^a^

Values are mean ± SD (*n* = 3). Means within columns with different letters (a,b) are significantly different (*p* < 0.05).

**Table 3 foods-11-00632-t003:** Effect of high-pressure processing (HPP), pasteurization (PT), and high-temperature short time (HTST) on color characteristics (L*, a*, b* and ΔE values) of passion fruit purée.

Treatments	Color Characteristics			
	L*	a*	b*	ΔE
Control	59.21 ± 1.25 ^b^	11.94 ± 0.32 ^a^	62.70 ± 0.45 ^ab^	0.00 ± 0.15 ^c^
HPP	60.78 ± 0.80 ^ab^	11.64 ± 0.40 ^a^	61.68 ± 0.91 ^b^	2.26 ± 0.06 ^b^
PT	61.92 ± 0.19 ^ab^	10.77 ± 0.65 ^a^	64.30 ± 0.30 ^a^	3.01 ± 0.39 ^a^
HTST	62.37 ± 0.36 ^a^	11.08 ± 0.67 ^a^	63.11 ± 0.13 ^ab^	3.06 ± 0.08 ^a^

Values are mean ± SD (*n* = 3). Means within columns with different letters (a–c) are significantly different (*p* < 0.05).

**Table 4 foods-11-00632-t004:** Effect of high-pressure processing (HPP), pasteurization (PT), and high-temperature short time (HTST) on antioxidants (TPC, TFC, Vitamin C, carotenoids, and TAC) of passion fruit purée.

Treatments	Control	HPP	PT	HTST
TPC(mg GAE/L)	1047.07 ± 18.67 ^a^	960.55 ± 16.20 ^bc^	1000.15 ± 45.63 ^ab^	907.76 ± 20.43 ^c^
TFC (mg RE/L)	172.98 ± 14.12 ^a^	189.78 ± 10.64 ^a^	168.11 ± 3.89 ^a^	132.40 ± 7.79 ^b^
Vitamin C (mg/mL)	251.68 ± 1.03 ^a^	238.94 ± 8.84 ^ab^	230.51 ± 9.02 ^ab^	223.80 ± 4.53 ^b^
AA (mg/mL)	122.26 ± 2.86 ^a^	113.54 ± 2.68 ^a^	70.65 ± 6.91 ^b^	59.36 ± 8.17 ^b^
DHAA (mg/mL)	129.42 ± 2.98 ^b^	125.40 ± 6.59 ^b^	159.86 ± 15.87 ^a^	164.44 ± 3.73 ^a^
β-carotene (µg/mL)	24.68 ± 1.57 ^a^	25.15 ± 3.57 ^a^	22.39 ± 1.73 ^ab^	16.30 ± 2.48 ^b^
Zeaxanthin (µg/mL)	25.55 ± 3.16 ^b^	40.05 ± 4.07 ^a^	28.87 ± 2.52 ^b^	25.88 ± 2.14 ^b^
β-cryptoxanthin (µg/mL)	45.36 ± 2.88 ^a^	27.55 ± 1.31 ^b^	10.78 ± 1.13 ^c^	9.19 ± 0.68 ^c^
Lycopene (µg/mL)	2.48 ± 0.21 ^a^	2.58 ± 0.08 ^a^	2.13 ± 0.21 ^a^	2.50 ± 0.12 ^a^
Total carotene (µg/mL)	98.07 ± 1.86 ^a^	95.33 ± 1.02 ^a^	64.17 ± 1.98 ^b^	53.88 ± 1.48 ^c^
DPPH (µmol/L)	4510.00 ± 149.00 ^a^	3533.33 ± 62.36 ^b^	3400.00 ± 177.95 ^b^	2900.00 ± 163.30 ^c^
ABTS^•+^ (µmol/L)	3275.15 ± 186.80 ^a^	2796.36 ± 262.64 ^b^	2523.64 ± 126.84 ^b^	2309.70 ± 66.94 ^c^

Values are mean ± SD (*n* = 3). Means within lines with different letters (a–c) are significantly different (*p* < 0.05).

## Data Availability

Data is contained within the article.

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
