# Peer review of "Comparison of the Effects of High Pressure Processing, Pasteurization and High Temperature Short Time on the Physicochemical Attributes, Nutritional Quality, Aroma Profile and Sensory Characteristics of Passion Fruit Purée"

_foods, 2022, doi:10.3390/foods11050632_

Round 1

Reviewer 1 Report

The authors did a very extensive and rather comprehensive study on the effect of pressure pasteurization and thermal pasteurization (here with two different time and temperature binomial combinations) on passion fruit puree.

did the authors evaluate what is the effect of the three applied pasteurization methods microorganisms inactivation? if yes they should give general picture have the results obtained, even if the results are published all will be published in another publication.

English language and style must be thoroughly improved.

please see my handwritten notes on the PDF file of the manuscript.

Overall, in my opinion, the manuscript needs to undergo revision to upgrade it to level to be possibly considered for a further processing in Foods.

Author Response

Dear reviewer,

Thank you for your careful work. We would like to express our great appreciation to you and reviewers for comments on our manuscript entitled “Comparison of the effects of high pressure processing, pasteurization and high temperature short time on the physicochemical attributes, nutritional quality, aroma profile and sensory characteristics of passion fruit purée” (foods-1536992).

We have studied comments carefully and made revision which marked in red in the manuscript. We have tried our best to revise our manuscript according to the comments. The responses to you are given as bellows.

Best regards,

Dr. Linyan Zhou   

*Corresponding author at: Faculty of Food Science and Engineering, Kunming University of Science and Technology, Kunming, Yunnan Province 650500, China, E-mail: [email protected]

Reviewer 2 Report

This article compares High Pressure Processing with control and 2 thermal treatments for passion fruit purée. The introduction is very well exposed. Materials and Methods accomplishes a hard experimental work and are also very well described. Results and discussion are reported in a correct format. One figure can be improved, because has very small letters.

In general, the whole article is very well organized and written. I only recommend some minor changes to improve it.

Author Response

(The authors gave the same response as above.)

Reviewer 3 Report

Comments to authors:

Major concerns:

This work is focuses on the effects of HPP (600 MPa/5 min), pasteurization (PT, 85 °C/30 s) and high temperature short time (HTST, 110 °C/8.6 s) on sensory related attributes (color, sugar, acid, aroma compounds), antioxidants (phenolic, vitamin C, carotenoids, antioxidant capacity) and sensory attributes of yellow passion fruit puree.

The manuscript is well organized, showing a great introduction section, well described methodology section and good discussion of the results. However, some points need to be reviewed and corrected. So, the manuscript needs a slight verification and correction.

Minor concerns:

  1. Line 23 – There is a sentence missing at the end of the abstract.
  2. Lines 95-96 – What was the pressurization and depressurization time?
  3. Page 8/At end of 1st paragraph – “Since it was found that polyphenol oxidase (PPO) and peroxidase (POD) in PFP were completely inactivated by thermal pasteurization in our preliminary experiments, it was deduced that the browning in PFP after PT and HTST treated was mainly induced by non-enzymatic browning, such as Maillard reaction, pigment destruction, and ascorbic acid oxidation.” - Support this information with literature.
  4. Figure 1 - Increase the size of the figures or separate them into 3 independent figures.
  5. Page 11/last two lines – “There were 4 carotenoids were detected in PFP, (..)” – delete the 2nd word “were”.

Author Response

(The authors gave the same response as above.)
